# Zygotic gene activation in the chicken occurs in two waves, the first involving only maternally derived genes

**Young Sun Hwang[1,2†], Minseok Seo[3,4†], Sang Kyung Kim[1,2], Sohyun Bang[3], Heebal Kim[1,2,3], Jae Yong Han[1,2]***

[1]Department of Agricultural Biotechnology, Seoul National University, Seoul, Republic of Korea; [2]Research Institute of Agriculture and Life Sciences, Seoul National University, Seoul, Republic of Korea; [3]C&K Genomics, Seoul, Republic of Korea; [4]Channing Division of Network Medicine, Brigham and Women's Hospital and Harvard Medical School, Boston, United States

**Abstract** The first wave of transcriptional activation occurs after fertilisation in a species-specific pattern. Despite its importance to initial embryonic development, the characteristics of transcription following fertilisation are poorly understood in Aves. Here, we report detailed insights into the onset of genome activation in chickens. We established that two waves of transcriptional activation occurred, one shortly after fertilisation and another at Eyal-Giladi and Kochav Stage V. We found 1544 single nucleotide polymorphisms across 424 transcripts derived from parents that were expressed in offspring during the early embryonic stages. Surprisingly, only the maternal genome was activated in the zygote, and the paternal genome remained silent until the second-wave, regardless of the presence of a paternal pronucleus or supernumerary sperm in the egg. The identified maternal genes involved in cleavage that were replaced by bi-allelic expression. The results demonstrate that only maternal alleles are activated in the chicken zygote upon fertilisation, which could be essential for early embryogenesis and evolutionary outcomes in birds.
DOI: https://doi.org/10.7554/eLife.39381.001

**\*For correspondence:**
jaehan@snu.ac.kr

[†]These authors contributed equally to this work

**Competing interests:** The authors declare that no competing interests exist.

## Introduction

The genetic events of early embryogenesis are initiated by zygotic genome activation (ZGA) (*Lee et al., 2014*; *Tadros and Lipshitz, 2009*). The timing and mechanisms of ZGA have been investigated in various species (*Aanes et al., 2011*; *Baugh et al., 2003*; *Harvey et al., 2013*; *Karr et al., 1985*; *Lee et al., 2013b*; *Leichsenring et al., 2013*; *Liang et al., 2008*; *Newport and Kirschner, 1982*; *Poccia et al., 1985*; *Tan et al., 2013*). In mammals, the first wave (1st wave) of transcriptional activation (also known as minor ZGA) occurs after fertilisation, during pronucleus (PN) formation. The subsequent second wave (2nd wave) of transcriptional activation (major ZGA) occurs during the two-cell stage of mice and the eight-cell stage of humans (*Aoki et al., 1997*; *Braude et al., 1988*; *Xue et al., 2013*). In avian species, reports in chicken and quail embryos have described gene activation during early cell cleavage (*Nagai et al., 2015*; *Olszańska et al., 1984*), but transcriptional activation has not been investigated during fertilisation. Recent studies suggest that there are two waves of ZGA in chickens based on mRNA profile (*Hwang et al., 2018aHwang et al., 2018c*). However, it is necessary to examine features such as de novo transcription in order to determine the timing and mechanisms of ZGA precisely.

The 1st wave of ZGA exhibits numerous characteristics that are species-dependent. In mice, the most distinctive feature of the 1st wave in the PN stage is that transcription from the paternal PN is greater than that from the maternal PN, due to the epigenetic regulation of the latter (*Aoki et al.,*

**eLife digest** The early stages of animal development involve a handover of genetic control. Initially, the egg cell is maintained by genetic information inherited from the mother, but soon after fertilization it starts to depend on its own genes instead. Activating genes inside the fertilized egg cell (zygote) so that they can take control of development is known as zygotic genome activation.

Despite the fact that birds are often used to study how embryos develop, zygotic genome activation in birds is not well understood. Fertilization in birds, including chickens, is different to mammals in that it requires multiple sperm to fertilize an egg cell. As such, zygotic genome activation in birds is likely to differ from that in mammals.

By examining gene expression in embryos from mixed-breed chickens, Hwang, Seo et al. showed that there are two stages of zygotic genome activation in chickens. The genes derived from the mother become active in the first stage, while genes from the father become active in the second stage. Genome activation in birds is therefore very different to the same process in mammals, which involves genome activation of both parents from the first stage. This extra level of control may help to prevent genetic complications resulting from the presence of multiple sperm, each of which carries a different set of genes from the father.

DOI: https://doi.org/10.7554/eLife.39381.002

*1997*; *Aoshima et al., 2015*; *Bouniol et al., 1995*; *Wu et al., 2016*; *Zhang et al., 2016*). In addition, the 1st wave is highly promiscuous, in that the expression of untranslatable mRNAs and intergenic regions is observed (*Abe et al., 2015*). In zebrafish, the mitochondrial genome is activated in the one-cell embryo (*Heyn et al., 2014*). In plants, the zygotic genome is activated soon after fertilisation, and rice zygotes show asymmetric activation of parental genomes (*Anderson et al., 2017*; *Chen et al., 2017*). As the earliest expressed genes in ZGA are species-specific (*Heyn et al., 2014*), the patterns of transcription during the 1st wave should be examined so that we can understand early embryogenesis in each species. However, no detailed investigation of transcription at fertilisation in avian species has been reported. As polyspermy is a distinctive feature in birds (*Snook et al., 2011*; *Iwao, 2012*), we hypothesised that the 1st wave derived from the parental genome would exhibit unique characteristics. Here, we conducted a genome-wide study of primary transcripts to clarify which genes undergo transcriptional activation during embryogenesis in chicken. We identified avian-specific expression patterns of the parental genome during the 1st wave. The results provide intriguing insights into initial the genome activation associated with physiological characteristics upon fertilisation in birds.

## Results and discussion

Detection of de novo transcription after fertilisation is difficult because of the large number of mRNAs that are being processed in the oocyte. We examined primary transcripts toassess the existence and timing of transcriptional activation accurately, using previously generated bulked embryonic whole-transcriptome sequencing (WTS) data (*Hwang et al., 2018aHwang et al., 2018c*) (*Figure 1A*). Hierarchical clustering of precursor mRNA (pre-mRNA) expression demonstrated that zygotes differed from oocytes, suggesting dynamic changes in primary transcripts after fertilisation (*Figure 1B*). Phosphorylated RNA polymerase II C-terminal domain first appeared during the late EGK.II to early EGK.III (*Nagai et al., 2015*), but the expression of pre-mRNA differed between EGK.III and EGK.VI (*Figure 1B*). The number of upregulated pre-mRNAs that are found in the zygote when compared to the oocyte provides evidence of a 1st wave (*Figure 1C*). In addition, a large number of pre-mRNAs were upregulated between EGK.III and EGK.VI, revealing the presence of a 2nd wave. This result is more direct evidence of the existence and timing of two waves of ZGA in chicken.

A number of expressed regions exhibited significant differences between the oocyte and zygote and between EGK.III and EGK.VI (*Figure 1—figure supplement 1*). The number of expressed regions was reduced during EGK.I and EGK.III but increased after EGK.VI. Of all of the genomic regions that are expressed, the proportion of expressed intronic regions decreased after fertilisation and increased gradually after EGK.VI (*Figure 1—figure supplement 2*). Unlike the expression

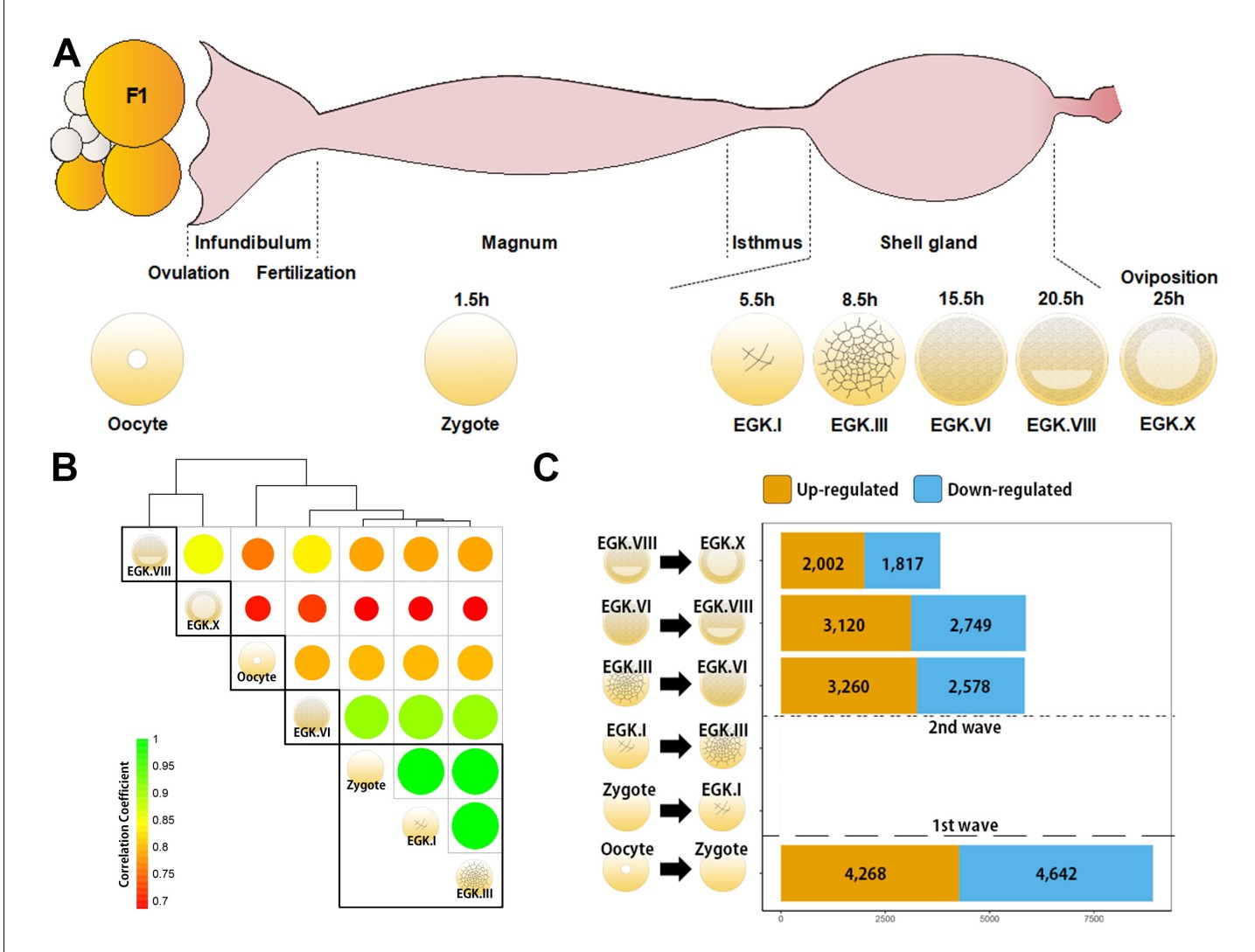

**Figure 1.** Genome-wide transcriptional activation during chicken early development. (**A**) Representative images of early embryos from oocyte to Eyal-Giladi and Kochav X (EGK.X) used for RNA-Seq and acquisition in the chicken oviduct. All embryos were classified following the morphological criteria of EGK. h, hours after fertilisation for each stage of embryos. (**B**) Hierarchical clustering of the whole transcriptome during early development in chicken. The size and colour of each circle represents the strength of the correlation coefficients based on whole-transcriptome expression. The black rectangle represents optimal clusters (k = 5) based on the Silhouette score. The transcriptomic changes between consecutive stages, including oocyte vs. zygote and EGK.III vs. EGK.VI, are shown. Zygote, EGK.I and EGK.III had similar transcriptome profiles. (**C**) Number of differentially expressed intronic regions in consecutive stages. The orange and blue colors represent up- and downregulated genes at 5% significance level after false discovery rate (FDR) multiple testing adjustment. The 1st wave of transcriptional activation between oocyte and zygote and the 2nd wave between EGK.III and EGK.VI are shown.

DOI: https://doi.org/10.7554/eLife.39381.003

The following figure supplements are available for figure 1:

**Figure supplement 1.** Quantification of the numbers of expressed regions including exons, introns and intergenic regions in the chicken genome.

DOI: https://doi.org/10.7554/eLife.39381.004

**Figure supplement 2.** Distribution of mapped reads on the exonic, intronic and intergenic regions during chicken early development.

DOI: https://doi.org/10.7554/eLife.39381.005

**Figure supplement 3.** Transcripts that undergo a detected change in expression between each stage during chicken early development.

DOI: https://doi.org/10.7554/eLife.39381.006

patterns seen during the minor ZGA in mammals (*Abe et al., 2015*), the proportion of expressed intergenic regions was constant regardless of transcriptional activation, indicating no expression of these regions during the 1st wave in chickens. In genic regions, large numbers of up- and downregulated mRNAs and long intergenic noncoding RNAs (lincRNAs) were observed during the 1st wave, while other RNAs were mostly downregulated after fertilisation (*Figure 1—figure supplement 3*), suggesting a potential role for long transcripts in the early cleavage stages. All RNA types were significantly upregulated during the 2nd wave.

We examined the candidate genes affected by the two waves using reverse transcription PCR (RT-PCR). Six upregulated genes in each wave were selected as representative genes (*Supplementary file 1*): *DLX6, GATA2, ZIC4, LYPD2, IFITM5* and *NKX6-3* for the 1st wave, and *WNT11, WNT3A, C8ORF22, NAT8L, PCOLCE2* and *AKAP2* for the 2nd wave. We successfully demonstrated two waves of transcriptional activation for all of the selected genes (*Figure 2* and *Figure 2—figure supplement 1*). The validated genes belonging to the 2nd wave of activation indicated a lack of transcriptional activity during rapid cellularisation in the cleavage period, and

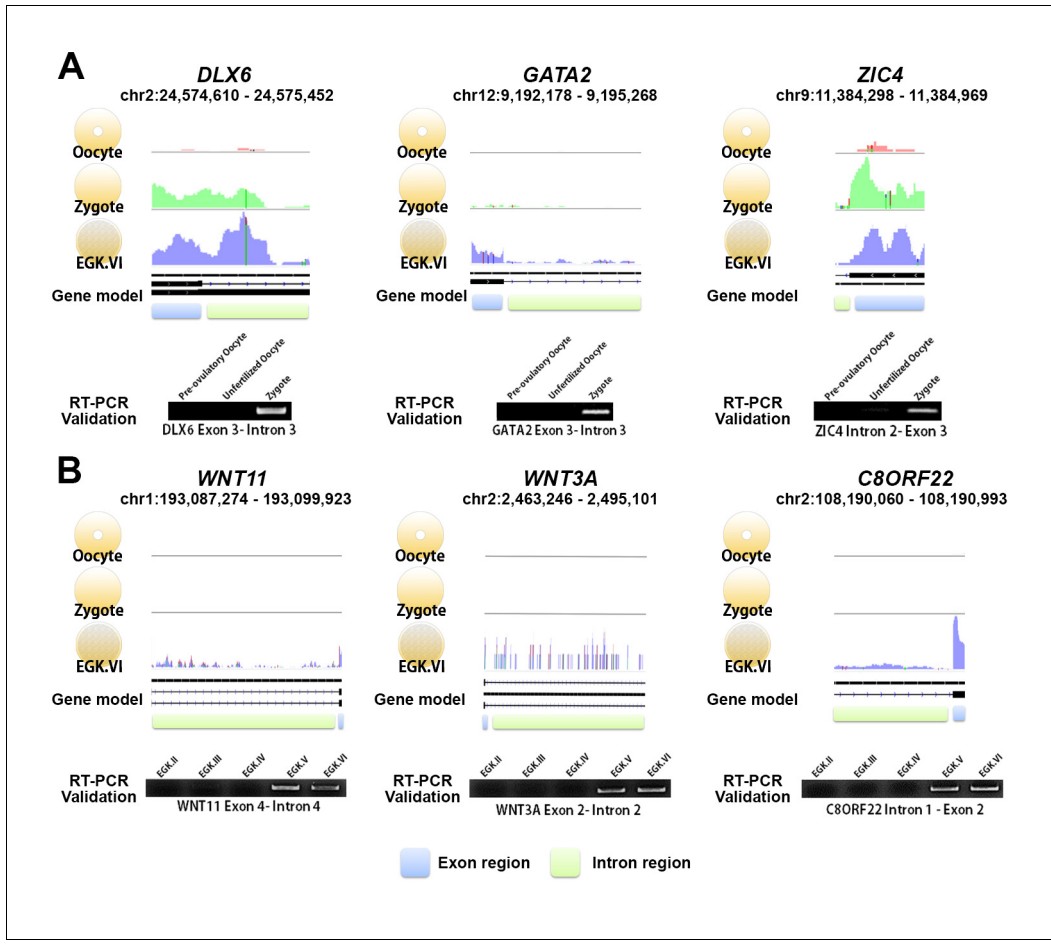

**Figure 2.** Exonic and intronic mapped reads on candidate genes related to the 1st and 2nd wave of transcriptional activation in chickens. (**A, B**) The pooled mapped reads based on the stage (three samples in each stage) were visualised using the Integrative Genomics Viewer tool. Detection with RT-PCR of gene activation via the appearance of primary transcripts based on whole-transcriptome sequencing and validation of the intronic expression of three genes (*DLX6, GATA2* and *ZIC4*) during the 1st wave (**A**) and of three different genes (*WNT11, WNT3A* and *C8ORF22*) during the 2nd wave (**B**). The following figure supplements are available for *Figure 2*.
DOI: https://doi.org/10.7554/eLife.39381.007

The following figure supplement is available for figure 2:

**Figure supplement 1.** Detection of gene activation and validation of intronic expression.
DOI: https://doi.org/10.7554/eLife.39381.008

showed that the 2nd wave of transcriptional activation in chicken occurred not between EGK.II and EGK.III, but between EGK.IV and EGK.V. The existence and timing of the two distinct waves of transcriptional activation were also confirmed experimentally and were consistent with the results of the bulked embryonic WTS analyses.

We hypothesised that the haploid nucleus of supernumerary sperm could be substantially induced during the 1st wave in addition to paternal and maternal PN activation because polyspermic fertilisation occurs in avian species. To assess this hypothesis, we generated multiomics data including whole-genome sequencing (WGS) and WTS. We completed WGS of six parents (three male Korean Oge (mKO) and three female White Leghorn (fWL) chickens) to identify breed-specific single-nucleotide polymorphisms (SNPs) (*Figure 3A*). We also generated single embryonic WTS data from hybrid oocyte, zygote and EGK.X blastoderms derived from the WGS-sequenced parents to examine the characteristics of the 1st wave-activated transcripts and of allelic expression. After confirming hybrid embryo formation between mKO and fWL (*Figure 3—figure supplement 1*), we collected oocytes, zygotes and EGK.X blastoderms from hens on the same day (*Figure 3—figure supplement 2*). Each embryo contained an average of 2.1 μg of total RNA (*Supplementary file 2A*). We performed the same analysis used in bulked embryonic WTS on single embryonic WTS to further establish the characteristics of the 1st wave. The WTS samples generated from the single embryos were clustered according to their respective stages (*Figure 3B*). A total of 4275 differentially expressed mRNAs were detected (*Figure 3C*; FDR-adjusted p<0.05), among which 1883 were upregulated and 2392 were downregulated in the zygote stage compared to the oocyte. We also observed that 118 and 786 lincRNAs were up- and downregulated, respectively. Owing to the dramatic changes in early development between fertilisation and oviposition, 10,298 mRNAs and 2507 lincRNAs were differentially expressed between the zygote and EGK.X stages (*Figure 3C*). We also observed a large number of primary transcripts that are upregulated in the zygote stage when compared to the oocyte stage(*Supplementary file 2B*; FDR-adjusted p<0.05). These results once again demonstrate that primary transcriptional activation occurs as developmental stage moves from oocytes to zygotes at single-embryo resolution, in terms of the numbers of differentially expressed pre-mRNAs and long transcripts.

Next, we identified parental allele-specific expression patterns during the 1st wave of transcriptional activation. A total of 1544 parentally derived SNPs were detected, distributed across 424 transcripts including mRNAs and lincRNAs (*Supplementary file 3A*). Interestingly, all of the transcripts that were identified in the zygote stage exhibited maternally derived expression during the 1st wave (*Figure 4A* and *Supplementary file 3A*). Most of the maternally derived transcripts, except for seven mRNAs and two lincRNAs, were replaced as bi-allelic expression occurred in the EGK.X stage. These nine transcripts could be interpreted as residual maternal transcripts that were not activated during the 2nd wave, rather than as genomic-imprinted genes, which are not conserved in avian species (*Frésard et al., 2014*). To verify this observation, we selected six pre-mRNAs (*MAP7D1*, *ESCO1*, *CCNB3*, *SYTL1*, *GRHL1* and *LLGL1*) that are upregulated during the 1st wave as representatives and validated the genotypes using Sanger sequencing (*Figure 4B* and *Supplementary file 3B*). All of the selected genes showed maternal allelic expression in the zygote until the EGK.VI stage, except for the *GRHL1* gene. These maternally derived genes converted to bi-allelic expression after the maternal-to-zygotic transition (MZT) at EGK.X. This phenomenon is distinguished from that in mammals, in which transcriptional activity in the paternal PN is two times greater than that in the maternal PN (*Aoki et al., 1997*). These results indicate that there is no possibility that the activated transcripts are derived from the supernumerary sperm nuclei and paternal PN, in contrast to the data from mammals (*Aoki et al., 1997*; *Bouniol et al., 1995*).

We examined the functional characteristics of the maternal genes that are activated during the 1st wave of transcriptional activation identified from the single embryonic WTS data. The analysis revealed that the 1st wave-activated maternal transcripts were enriched in the following pathways: cell cycle; Notch signalling pathway; Wnt signalling pathway; regulation of transcription, DNA-templated; and regulation of small GTPase-mediated signal transduction (*Figure 4—figure supplement 1* and *Supplementary file 4*). These pathways were activated from the maternal genome and are involved in rapid asymmetric cellularisation during the cleavage period in chickens (*Hwang et al., 2018c*) and other species (*Castanon et al., 2013*; *Huang et al., 2015*; *Priess, 2005*; *Tse et al., 2012*; *Zhang et al., 2014*). While the 1st wave in mice promotes the low-level expression of

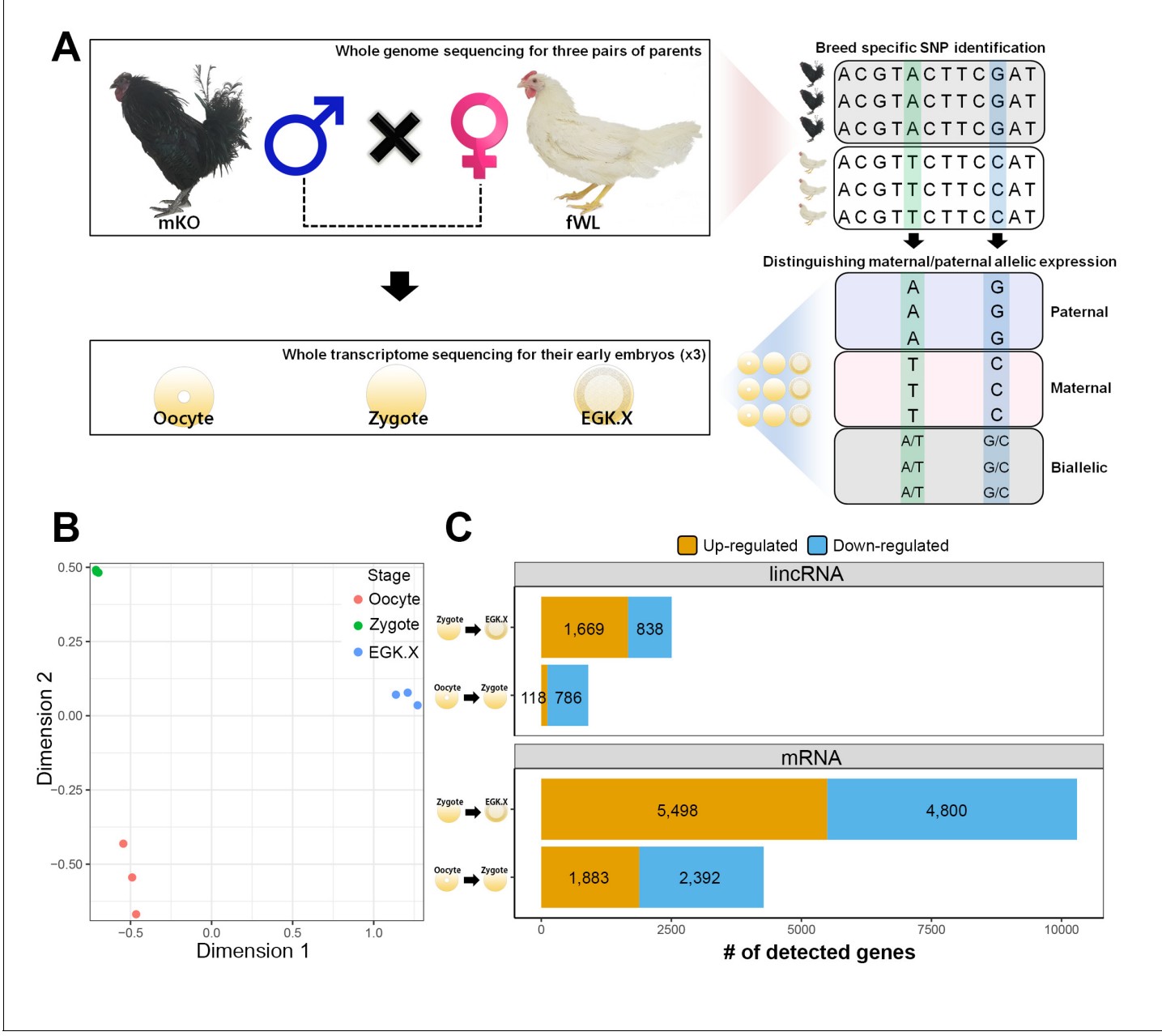

**Figure 3.** Whole-transcriptome analysis of single early chicken embryos. (**A**) Schematic diagram of the experimental design using a multiomics approach to assess allelic expression. Three pairs of parental male Korean Oge (mKO) and female White Leghorn (fWL) chickens were subjected to whole-genome sequencing. Hybrid single embryos between mKO and fWL at the oocyte, zygote and EGK.X stages from each parent were subjected to whole-transcriptome sequencing. Allelic expression in the hybrid embryos was examined on the basis of breed-specific SNPs. (**B**) Multidimensional scaling (MDS) plot based on log2 trimmed mean of M-value (TMM) normalised gene expression of the whole transcriptome in pre-oviposited chicken embryos. Biological triplicates of single embryos were clustered, and three developmental stages were distinct. (**C**) Number of significantly detected long transcripts (mRNAs and lincRNAs) detected by comparing gene expression among single oocytes, zygotes and EGK.X embryos (FDR-adjusted p<0.05).

DOI: https://doi.org/10.7554/eLife.39381.009

The following figure supplements are available for figure 3:

**Figure supplement 1.** Confirmation of hybrid embryos (Hamburger and Hamilton stage 4) from crosses between female White Leghorn (fWL) and male Korean Oge (mKO) using breed-specific primers.

DOI: https://doi.org/10.7554/eLife.39381.010

**Figure supplement 2.** Schematic diagram of single oocyte, zygote and EGK.X embryo acquisition from one hen on the same day.

*Figure 3 continued on next page*

Figure 3 continued

DOI: https://doi.org/10.7554/eLife.39381.011

numerous non-functional genes (*Abe et al., 2015*), the maternal genes activated during the 1st wave in chickens seem to be related to early cell division in embryogenesis.

As demonstrated in previous studies, the characteristics of the 1st wave vary among species (*Abe et al., 2015*; *Anderson et al., 2017*; *Chen et al., 2017*; *Heyn et al., 2014*). Our results suggest the exclusive activation of maternal alleles after fertilisation in chicken (*Figure 4C*). However, after

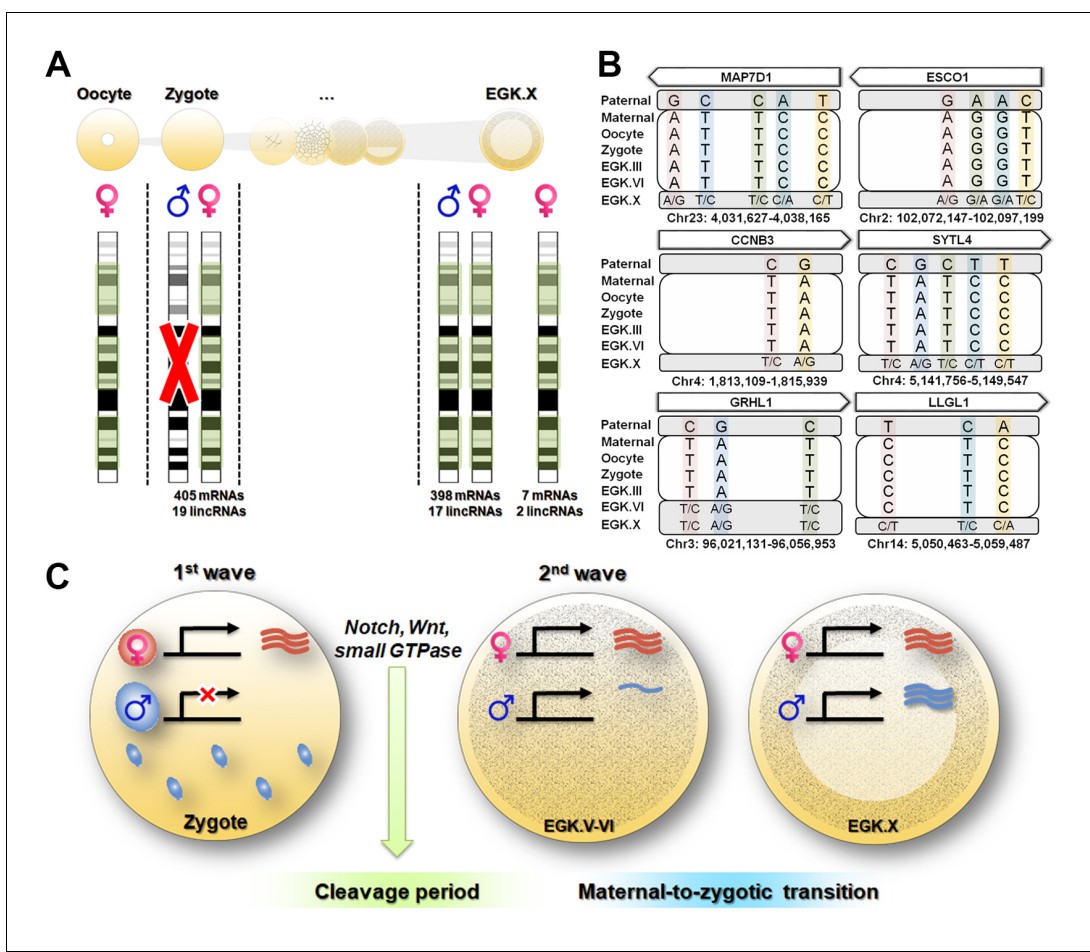

**Figure 4.** Maternal genome activation (MGA) during the 1st wave of transcriptional activation in chicken zygote. (**A**) Determination of parental allelic expression from the zygote stage. Only maternal alleles were observed in transcripts induced by st st activation. These maternally derived upregulated genes showed bi-allelic expression after EGK.X. (**B**) Validation of 1st wave transcription-induced maternal allelic expression by Sanger sequencing. The maternal transcription profile after the 1st wave changed to bi-allelic expression between EGK.VI and EGK.X after the 2nd activation. (**C**) Schematic summary of genome activation during chicken early development. Only MGA occurred after fertilisation and this wave of gene activation may regulate the cleavage period.
DOI: https://doi.org/10.7554/eLife.39381.012

The following figure supplements are available for figure 4:

**Figure supplement 1.** Functional classification of genes by maternal genome activation during the 1st wave of transcriptional activation and tracing through early development.
DOI: https://doi.org/10.7554/eLife.39381.013

**Figure supplement 2.** Hypothetical diagram for avian polyspermy and only maternal genome activation after fertilisation.
DOI: https://doi.org/10.7554/eLife.39381.014

MZT, most expressed genes were derived from both paternal and maternal genomes. Functionally, transcripts affected by the 1st wave were involved in asymmetric rapid cellularisation and in the fundamental regulation of further development (*Figure 4C*). We speculate that this evolved by necessity in animals following physiological polyspermy (*Figure 4—figure supplement 2*). Polyspermic animals require a number of sperm to activate large eggs (*Iwao, 2012*). In addition to pathological mitosis (*Snook et al., 2011*), polyspermic embryos of sea urchin demonstrated that transcriptional activation after fertilisation was greatly stimulated by the PN of supernumerary sperm (*Poccia et al., 1985*). Such a disproportionate genome contribution could result in an excessive amount of transcription. The total polyspermy number reportedly varies (*Hemmings and Birkhead, 2015*; *Lee et al., 2013a*) and is positively correlated with egg size (*Birkhead et al., 1994*). Individual sperm provide genomic diversity (*Wang et al., 2012*) but could result in genomic instability if different types of transcripts are expressed by various sperm nuclei. Therefore, polyspermic animals may have evolved means of inhibiting the activation of the paternal PN to control gene expression levels from the 1st wave. Taken together, our results suggest that the maternally derived 1st wave is essential for early development and evolutionary outcomes in avian species.

# Materials and methods

**Key resources table**

| Reagent type (species) or resource | Designation | Source or reference | Identifiers | Additional information |
| --- | --- | --- | --- | --- |
| Sequence-based reagent | Breed-specific primers | (*Choi et al., 2007*) | | See elsewhere in 'Materials and methods' |
| Sequence-based reagent | RT-PCR primers | This paper | | See **Supplementary file 6** |
| Commercial assay or kit | DNeasy Mini Kit | Qiagen | Qiagen:69504 | |
| Commercial assay or kit | TRIzol reagent | Invitrogen | Invtirogen:15596026 | |
| Commercial assay or kit | SuperScript III First-Strand Synthesis System | Invitrogen | Invitrogen:18080051 | |
| Commercial assay or kit | pGEM-T Easy Vector Systems | Promega | Promega:A1360 | |
| Software, algorithm | Code used for RNA-seq quantification analysis | This paper | | The python code used for RNA-seq quantification analysis. See **Source code 1** |

## Experimental animals and animal care

The experimental use of chickens was approved by the Institute of Laboratory Animal Resources, Seoul National University (SNU-150827–1). The experimental animals were cared for according to a standard management program at the University Animal Farm, Seoul National University, Korea. The procedures for animal management, reproduction and embryo manipulation adhered to the standard operating protocols of our laboratory.

## Identification of differentially expressed regions during early developmental stages of chickens

To detect de novo transcription, the analytical approach to primary transcripts used in previous studies of other species (*Abe et al., 2015*; *Graf et al., 2014*; *Lee et al., 2013b*; *Paranjpe et al., 2013*) was followed. In the quantification step, four types of genomic regions were considered: transcripts, exons, introns and intergenic regions. Although quantification of the transcript and exon level can be achieved directly without any pre-processing steps by using the galGal4 gene annotation file (GTF), the genomic position needs to be defined in order to estimate the expression levels of the intron and intergenic regions. When defining intron area, overlapping annotation of the exon within the associated gene makes it difficult to define intron regions from the reference genome. In addition, information from different strands should be considered when defining intron regions between each exon. To address these issues, intron region was defined using custom python script

(*Source code 1*). As in the method used to define the intron region between exons within the associated gene, python script was used to define intergenic regions between genes within the same chromosome. After defining intronic and intergenic regions, a GTF was generated using the coordinate information. Expression levels were measured with HTSeq-count (v 0.6.1) on the basis of the the GTFs (*Anders et al., 2015b*).

To explore gene expression changes during early developmental stages, pre-existing bulked embryonic WTS data covering the oocyte, zygote, EGK.I, EGK.III, EGE.VI, EGK.VIII and EGK.X stages (GSE86592) (*Hwang et al., 2018bHwang et al., 2018c*) were employed. Three types of matrix data were generated, and these data were employed in statistical analyses. Six statistical tests, oocyte vs. zygote, zygote vs. EGK.I, EGK.I vs. EGK.III, EGK.III vs. EGK.VI, EGK.VI vs. EGK.VIII and EGK.VIII vs. EGK.X, were performed using the edgeR package (*Robinson et al., 2010*) in the matrix data derived from intron and intergenic regions separately. More detailed contrast tests were performed on the generalised linear model. In this study, a result was considered significant at a FDR-adjusted p-value of $p<0.05$ (*Benjamini and Hochberg, 1995*).

## Genomic DNA isolation and DNA sequencing library preparation for WGS data

Genomic DNA was isolated from blood collected from the wing vein of six parental chickens (three mKO and three fWL) using 1 mL 30-gauge syringes (Shina Corporation, Seoul, Korea). The blood samples were transferred into EDTA tubes (BD Biosciences, San Jose, CA, USA) immediately after collection. Blood (10 μL) was used for isolation of genomic DNA using a DNeasy Mini Kit (Qiagen, Valencia, CA, USA). The quality of the extracted genomic DNA was determined using the Trinean DropSense96 system (Trinean, Gentbrugge, Belgium), RiboGreen (Invitrogen, Carlsbad, CA, USA) and an Agilent 2100 Bioanalyzer (Agilent Technologies, Santa Clara, CA, USA). Genomic DNA was used for the construction of cDNA libraries using a TruSeq Nano DNA LT Library Preparation Kit (Illumina Inc., San Diego, CA, USA). The resulting libraries were subjected to chicken genome resequencing (30 × coverage) using the Illumina Nextseq 500 platform to produce paired 150 bp reads. The raw sequencing data were deposited in BioProject under accession number PRJNA393895.

## RT-PCR for confirmation of hybrid embryos

Before collecting early embryos, EGK.X blastoderms formed from crosses between mKO and fWL were incubated in a chamber at 37.5°C under 80% humidity for 18 hr. Genomic DNA was isolated from Hamburger and Hamilton stage 4 (HH4) (*Hamburger and Hamilton, 1951*) embryos using a DNeasy Mini Kit (Qiagen). RT-PCR was performed to confirm hybridisation between KO and WL using breed-specific primers (AS3554-I9/P5FWD WL F: 5′-AGC AGC GGC GAT GAG CGG TG-3′; WL R: 5′-CTG CCT CAA CGT CTC GTT GGC-3′; AS3554-WT/P5FWD KO F: 5′-AGC AGC GGC GAT GAG CAG CA-3′; KO R: 5′-CTG CCT CAA CGT CTC GTT GGC-3′) (*Choi et al., 2007*), with an initial incubation at 95°C for 10 min, followed by 35 cycles of 95°C for 30 s, 69°C for 30 s and 72°C for 30 s. The reaction was terminated after a final incubation at 72°C for 10 min.

## Alignment and variant calling for WGS data

The paired-end reads for six chickens (three biological replications of mKO and fWL breeds) were generated using the Illumina Nextseq 500 platform. In total, 8.38 billion reads or ~2.53 Gbp of sequences were generated. Paired-read sequences were selected for quality using Trimmomatic (v0.33) (*Bolger et al., 2014*). Using Bowtie 2 (v2.2.5) (*Langmead and Salzberg, 2012*), reads were aligned to the reference genome sequence galGal4 (Build v 4.82) with an average alignment rate of 91.61%. After potential PCR duplicates were filtered and misalignments resulting from the presence of insertions and deletions (INDELs) were corrected, SNPs were detected using GATK v3.4.46 (*McKenna et al., 2010*). More detailed, potential PCR duplicates were filtered using the option 'REMOVE_DUPLICATES = true' in the 'MarkDuplicates' open-source tool of Picard (v 1.138) (https://broadinstitute.github.io/picard/). SAMtools (v1.2) (*Li et al., 2009*) was then employed to create index files for reference and Binary Alignment/Map (BAM) files. In the variant-calling step with GATK v3.1, local realignment of reads to correct misalignments was performed because of the presence of INDELs ('*RealignerTargetCreator*' and '*IndelRealigner*' arguments). In the GATK tool, two types of arguments, '*UnifiedGenotyper*' and '*SelectVariants*' were employed for variant calling. In addition,

'*VariantFiltration*' was applied to filter bad variants on the basis of the following criteria: (1) variants with a Phred-scaled quality score <30 were filtered; (2) SNPs with '*mapping quality zero (MQ0) >4*', '*quality depth <5*' and '*(MQ0 / (1.0*DP))>0.1*' were filtered; and (3) SNPs with '*Phred-scaled P value using Fisher's exact test >200*' were filtered. As a result, 10,529,469 variants were detected, of which 9,805,997 variants (93.129%) were previously known variants (*Supplementary file 5A, B*).

## Chicken early hybrid embryo preparation, RNA isolation and library preparation for single embryonic WTS data

The egg-laying times of three fWLs, which were mated with mKOs, were recorded. A single hybrid EGK.X blastoderm was collected from WL hens after oviposition. To collect single oocytes and hybrid zygotes, WL hens were sacrificed and their follicles were harvested. Oocytes and hybrid zygotes were collected simultaneously from one WL hen. Owing to the small transcriptomic differences between pre- and post-ovulatory oocytes observed in the previous study (*Elis et al., 2008*) and the infeasibility of simultaneous acquisition of post-ovulatory oocytes and zygotes from a single hen, only the pre-ovulatory large F1 oocyte was isolated. Only zygote embryos not showing cleavage and located in the magnum were collected within 1.5 hr after fertilisation, according to the recorded egg-laying times (*Figure 3—figure supplement 2*). All embryos were classified according to morphological criteria (*Figure 1A*). Shortly after collection, the embryos were separated from the egg using sterile paper, and the shell membrane and albumen were detached from the yolk. A piece of filter paper (Whatman, Maidstone, UK) with a hole in the centre was placed over the germinal disc. After cutting around the paper containing the embryo, it was gently turned over and transferred to saline to further remove the yolk and vitelline membrane to allow embryo collection. Total RNA was isolated from early embryos using TRIzol reagent (Invitrogen). The quality of the extracted total RNA was determined using the Trinean DropSense96 system (Trinean), RiboGreen (Invitrogen) and an Agilent 2100 Bioanalyzer (Agilent Technologies). Total RNA was used for construction of cDNA libraries using a TruSeq Stranded Total RNA Sample Preparation Kit (Illumina, Inc.). The resulting libraries were subjected to whole-transcriptome analysis using the Illumina Nextseq 500 platform to produce paired 150 bp reads. The raw sequencing data were deposited in Gene Expression Omnibus (GEO) under accession number GSE100798.

## Quality control, alignment and quantification of mapped reads for single embryonic WTS data

Trimmomatic (v 0.33) (*Bolger et al., 2014*) was used to generate clean reads. Per-base sequence qualities were checked using FastQC (v 0.11.2) (*Andrews, 2010*) and filtered fastq files. Trimmed reads were aligned to the galGal4 genome files using the HISAT2 alignment software (v 2.0.0) (*Kim et al., 2015*) with the following alignment option: '*–rna-strandness RF*'. Sequence Alignment/ Map (SAM) files were converted into compressed and sorted BAM files using SAMtools (v 1.4.1) (*Li et al., 2009*). The mapped reads were quantified using HTSeq-count (*Anders et al., 2015a*) with the merged GTF, with total RNAs and lincRNAs derived from Ensembl and ALDB (*Li et al., 2015*), respectively. The quantification of mapped reads on intronic regions for single embryonic WTS data was performed using the procedure also used for bulked embryonic WTS data.

### Variant calling RNA-Seq

Using the alignment file (.BAM), potential PCR duplicates were removed using the Picard (v 1.138) software with '*REMOVE_DUPLICATES = true*' in the '*MarkDuplicates*' option. After that, the SplitN-CigarReads tool implemented in GATK was performed with the '*-rf ReassignOneMappingQuality -RMQF 255 -RMQT 60 U ALLOW_N_CIGAR_READS*' option. In the variant-calling step with GATK, local realignment of reads was performed to correct misalignments (using the '*RealignerTargetCreator*' and '*IndelRealigner*' options). Finally, base-recalibration was performed using BaseRecalibrator implemented in GATK with known variant sites in galGal4. Using HaplotypeCaller in the GATK tool, variant calling was performed with the '*-dontUseSoftClipped-Bases -stand_call_conf 20.0 -stand_emit_conf 20.0*' option. Finally, bad variants were filtered using the VariantFiltration tool with '*-window 35 -cluster 3 -filterName FS -filter 'FS >30.0' -filterName QD -filter 'QD <2.0'*' option. At the end of this process, 265,788 variants were detected, of which 248,030 variants (93.319%) were previously known sites (*Supplementary file 5C, D*).

## Identification of the maternally and paternally expressed genes through detection of breed-specific variants

Maternal and paternal samples were genotyped using WGS, and their offspring, including maternal oocytes, were genotyped using WTS (variant calling on the RNA-Seq data). After pre-processing, there were two types of genotype data (DNA and RNA sequencing data) available for the mother, father, oocyte, zygote and EGK.X. In two types of SNP data, 10,529,469 and 265,788 variants were detected in DNA and RNA sequencing data, respectively. First, breed-specific SNPs (such as , first, SNPs '0/0' and '1/1' genotype for maternal and paternal groups, respectively; and second, SNPs '1/1' and '0/0' genotype for maternal and paternal groups, respectively) were identified and annotated using SnpSift (*Cingolani et al., 2012*) in parental SNP data. As a result, 216,003 SNPs were identified as breed-specific SNPs. After that, two SNP datasets (breed-specific SNPs and their offspring genotypes derived from the RNA-Seq data) were combined to detect maternally and paternally expressed genes, and 14,817 SNPs were commonly identified in breed-specific SNPs and those derived from RNA-Seq data. Using these combined genotype data, three types of filtering steps were carried out. First, mismatched genotypes of the reference and alternative allele between breed-specific SNPs and SNPs derived from the RNA-Seq were removed; two variants were removed in this step. Second, different genotypes within the biological replicates were removed; 9,143 SNPs were removed in this step. Finally, mismatched genotypes between maternal samples and oocyte samples were removed; six SNPs were removed in this step. The remaining 5,666 SNPs were annotated using the SnpSift tool with galGal4 and ALDB GTFs. To find the most conservative evidence of parental expression, if a single SNP was found within the gene or genotype pattern that was not consistent among the SNPs, it was filtered out. In addition, unannotated SNPs in both databases, Ensembl and ALDB, were removed to facilitate biological interpretation. At the end of this process, 1,544 SNPs were detected as parental expression markers, all of which showed a maternal expression pattern (*Supplementary file 5E*).

## Identification of functional characteristics of differentially expressed genes

On the basis of the biological process terms (BP terms) of the GO and KEGG pathways, functional enrichment tests using DAVID (*Dennis et al., 2003*) were performed on the differentially expressed genes.

## Exon–intron RT-PCR and validation of allelic expression

Total RNA (1 µg) was used as the template for cDNA synthesis using the SuperScript III First-Strand Synthesis System (Invitrogen). The cDNA was serially diluted 5-fold and equalised quantitatively for PCR amplification. To validate allelic expression, additional single hybrid embryos at EGK.III and VI were collected from parents with identical genotypes as confirmed by WGS, and their total RNA isolation and cDNA synthesis were performed as described above. Primers for exon–intron PCR of 12 genes and for allelic expression of six genes were designed using the program Primer3 (*Untergasser et al., 2012*) (*Supplementary file 6A, B*). RT-PCR was performed with an initial incubation at 95°C for 5 min, followed by 35 cycles of 95°C for 30 s, 59°C for 30 s and 72°C for 30 s. The reaction was terminated after a final incubation at 72°C for 5 min. PCR products were cloned into the pGEM-T Easy Vector (Promega, Madison, WI, USA) for sequencing with an ABI 3730xl DNA Analyzer (Applied Biosystems, Foster City, CA, USA).

## Additional information

### Funding

| Funder | Grant reference number | Author |
| --- | --- | --- |
| National Research Foundation of Korea | NRF-2015R1A3A2033826 | Jae Yong Han |

The funders had no role in study design, data collection and interpretation, or the decision to submit the work for publication.

## Author contributions
Young Sun Hwang, Conceptualization, Data curation, Formal analysis, Validation, Investigation, Visualization, Methodology, Writing—original draft, Writing—review and editing; Minseok Seo, Data curation, Software, Formal analysis, Validation, Visualization, Methodology, Writing—original draft, Writing—review and editing; Sang Kyung Kim, Investigation, Visualization, Methodology; Sohyun Bang, Software, Methodology; Heebal Kim, Conceptualization, Resources, Software, Supervision, Writing—original draft, Writing—review and editing; Jae Yong Han, Conceptualization, Resources, Supervision, Funding acquisition, Writing—original draft, Project administration, Writing—review and editing

## Author ORCIDs
Young Sun Hwang (iD) http://orcid.org/0000-0002-6443-9579
Minseok Seo (iD) http://orcid.org/0000-0002-5364-7524
Sohyun Bang (iD) http://orcid.org/0000-0003-2058-1079
Jae Yong Han (iD) http://orcid.org/0000-0003-3413-3277

## Ethics
Animal experimentation: The experimental use of chickens was approved by the Institute of Laboratory Animal Resources, Seoul National University (SNU-150827-1). The experimental animals were cared for according to a standard management program at the University Animal Farm, Seoul National University, Korea. The procedures for animal management, reproduction and embryo manipulation adhered to the standard operating protocols of our laboratory.

## Decision letter and Author response
Decision letter https://doi.org/10.7554/eLife.39381.030
Author response https://doi.org/10.7554/eLife.39381.031

# Additional files

## Supplementary files
• Supplementary file 1. Gene list and expression of transcripts for exon–intron PCR.
DOI: https://doi.org/10.7554/eLife.39381.015

• Supplementary file 2.  (**A**) Total RNA quantity of a single chicken early embryo. (**B**) Upregulated intronic expression between single oocyte and zygote (FDR-adjusted $p<0.05$).
DOI: https://doi.org/10.7554/eLife.39381.016

• Supplementary file 3.  (**A**) Variant calling of single hybrid embryo RNA-Seq to determine which parental allele was expressed. (**B**) Gene list and expression of genotyped transcripts by Sanger sequencing.
DOI: https://doi.org/10.7554/eLife.39381.017

• Supplementary file 4. Significantly detected biological processes of GO and KEGG pathways on the basis of upregulated DEGs between single oocytes and zygotes.
DOI: https://doi.org/10.7554/eLife.39381.018

• Supplementary file 5.  (**A**) Detected SNPs on each chromosome from the WGS data. (**B**) Quality information for detected SNPs in WGS data. (**C**) Detected SNPs on each chromosome from the WTS data. (**D**) Quality information for detected SNPs in WTS data. (**E**) Detected maternal SNPs in multiomics analysis.
DOI: https://doi.org/10.7554/eLife.39381.019

• Supplementary file 6.  (**A**) Primers used for the exon–intron RT-PCR. (**B**) Primers used for the validation of allelic expression.
DOI: https://doi.org/10.7554/eLife.39381.020

• Source code 1. Python script for generating intron and intergenic regions based on the Ensembl GTF.

DOI: https://doi.org/10.7554/eLife.39381.021
• Transparent reporting form
DOI: https://doi.org/10.7554/eLife.39381.022

## Data availability

Generated WGS of parental chickens has been deposited in BioProject under accession number PRJNA393895 (https://www.ncbi.nlm.nih.gov/bioproject/?term=PRJNA393895). Generated single hybrid embryonic WTS data has been deposited in GEO under accession number GSE100798 (https://www.ncbi.nlm.nih.gov/geo/query/acc.cgi?acc=GSE100798). Published bulked embryonic WTS data are available under accession number GSE86592 (https://www.ncbi.nlm.nih.gov/geo/query/acc.cgi?acc=GSE86592).

The following datasets were generated:

| Author(s) | Year | Dataset title | Dataset URL | Database and Identifier |
|---|---|---|---|---|
| Han J, Hwang Y | 2018 | Avian zygote activates only maternal allele to disburden high variation of supernumerary sperms contrary to mammal | https://www.ncbi.nlm.nih.gov/geo/query/acc.cgi?acc=GSE100798 | NCBI Gene Expression Omnibus, GSE100798 |
| Han J, Hwang Y | 2018 | Avian zygote activates only maternal allele to disburden high variation of supernumerary sperms contrary to mammal | https://www.ncbi.nlm.nih.gov/bioproject/?term=PRJNA393895 | NCBI BioProject, PRJNA393895 |

The following previously published datasets were used:

| Author(s) | Year | Dataset title | Dataset URL | Database and Identifier |
|---|---|---|---|---|
| Han JY | 2017 | Developmental programs in chicken early embryos by whole transcriptome analysis | https://www.ncbi.nlm.nih.gov/geo/query/acc.cgi?acc=GSE86592 | NCBI Gene Expression Omnibus, GSE86592 |

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
