## [Decision Letter]

[Editors’ note: a previous version of this study was rejected after peer review, but the authors submitted for reconsideration. The first decision letter after peer review is shown below.]

Thank you for submitting your work entitled "The only maternal genome is activated in avian zygotes after fertilization" for consideration by *eLife*. Your article has been reviewed by three peer reviewers, and the evaluation has been overseen by a Reviewing Editor and a Senior Editor. The following individual involved in review of your submission has agreed to reveal their identity: Claudio D Stern (Reviewer #1).

Our decision has been reached after consultation between the reviewers. Based on these discussions and the individual reviews below, we regret to inform you that your work will not be considered further for publication in *eLife* at this time.

As you will see in the reviews below, the reviewers found many positive aspects of the study and think it may still be suitable for publication in *eLife*; however, ambiguity about which findings are novel here relative to other recent papers must be addressed before this work can be considered further. For example, you'll see one of the reviewers interpreted the two waves of zygotic activation as being reported for the first in this study. I appreciate that you have delineated the differences among your recent papers to the *eLife* staff, but the paper itself will also need to be substantially revised to more clearly focus only on the novel findings that are uniquely presented here. The reviewers need to see this revised version before they can assess suitability for *eLife*, thus we are rejecting this work now, but leaving the door open for resubmission of a revised version as a new manuscript. This revision must also address the other concerns raised in the reviews below, including verification of the sequence analysis pipeline. The manuscript should also be carefully evaluated for proper English, as grammatical problems make it hard to understand in some places; even the title is confusing as written.

Reviewer #1:

This is potentially an important paper exploring the timing of first activation of the zygotic genome in an avian embryo, the chick (*Gallus gallus*). Using a carefully designed set of experiments (mainly transcriptomics, also taking advantage of crosses between two different strains of parents with distinctive SNPs), the authors reveal that there are two separate waves of zygotic gene activation (ZGA): a first major wave occurring soon after fertilization, in which only genes located in maternally-derived chromosomes are activated, and this is followed some hours later (stage EGK V) where genes on both maternally- and paternally-derived chromosomes become active.

Overall I think that this is a very carefully conducted study and that it reveals very important information which was lacking until now about when the zygotic genome is activated in the chick embryo, an important model system. A peculiarity of avian species is that fertilization is highly polyspermic and the authors speculate that these two waves of genome activation, where the paternally-derived genes are activated comparatively late, is a consequence of polyspermy. In anamniotes (including invertebrates as well as Amphibians and fishes), the zygotic genome is only activated late, after about 10 synchronous cell divisions (at the mid-blastula stage, a process generally known as the mid-blastula transition MBT); therefore any patterning and cell commitment events until this time must depend on inherited maternal determinants since there cannot be differential gene expression before MBT. This paper is important because it shows that, as in mammals, the bird genome is indeed activated very early (but with the peculiarity that only maternally-derived genes are turned on first), which allows differential gene expression almost from the start. Interestingly, among the genes/pathways enriched in the first wave of zygotic gene expression, are genes involved in intercellular cell signalling, especially Notch and Wnt pathways. This suggests again that early patterning events in the amniote embryo rely on zygotic, rather than maternal gene products.

I only have one major comment: the paper is very poorly written and the English needs a lot of attention. In places it is almost impossible to understand (even the title makes no sense, as are parts of the Abstract). It is essential that the authors seek help from native English speakers who could help to make this paper readable. I find myself unable to criticise it more deeply because there are several parts which I don't understand.

When re-writing, please be careful to distinguish very clearly between transcription from "maternally-derived genes", and "maternal determinants" (i.e. gene products inherited from the mother, as in anamniotes before ZGA) – because the concept of MBT and late activation of the zygotic genome in many model species is so engrained in the developmental literature, there is some danger than some readers may be confused by these to quite different concepts/findings.

Reviewer #2:

In this manuscript the authors analyse the transcriptome of chicken embryos at pre-oviposition stages. Whole genome RNA-seq analysis was carried out on mature follicles and early EGK stages. The authors found abundant maternal RNAs and increases in transcripts at two separate developmental stages: the zygote and EG V. In previous work by these authors, they used the RNA from these developmental stages to define expression clusters that changed during early uterine development (Hwang, 2018). In this manuscript the authors used interbreed crosses to assess paternal and maternal transcripts. They found that maternal transcripts only increased after the oocyte state and conclude that the paternal genome is not activated at early stages. The authors hypothesise that polyspermy in bird species has led to the transcription silencing of the paternal genome during early development. The hypothesis is intriguing but the paper needs revision and statistical analysis to determine if the hypotheses are proven.

1) Not enough information is given on the bioinformatics analysis of variants presented for the data in Supplementary file 5E. Allele depth, read depth and SNP quality needs to be presented.

2) How are the 8 gene expression clusters identified in the author's previous paper (Huang, 2018) related to the DEGs identified in this paper?

3) My concern is the ability to distinguish between maternal transcripts in the early zygote and de novo RNA transcription. This is premised by the author's statement:

'We also investigated primary transcripts throughout early development to obtain definitive evidence of gene activation as the massive alteration of maternally stored RNAs after fertilization may have masked the smaller effects of the 1st activation.'

The presence of maternal RNAs is clouding the analysis of de novo transcription. Can the authors identify maternal transcripts through looking at unmapped reads from the 3' end of follicle transcripts and either de novo transcript assembly on these follow by a molecular biology analysis of the transcripts to verify the maternal 3' UTR patterns?

4) I believe the title is miswritten. 'Only the maternal genome is transcribed in avian zygotes after fertilisation'.

5) There are several problems with nomenclature. Referring to avian zygotes suggests that several avian species were examined. The authors only examined chicken (*Gallus gallus*). We do not know if this mechanism is conserved in other bird species.

6) The English needs to be improved. Sections of the paper are difficult to understand.

7) Figure 1A: A time scale is needed. Chicken oviposition requires ~20-24 hours. An approximate stages of EGK should be given.

*Reviewer #3:*

The authors Hwang et al., have submitted a manuscript entitled 'The only maternal genome is activated in avian zygotes after fertilization'. In regards to the mechanisms of the zygotic genome activation, very few reports are available in avian species comparatively to the mammalian ones. This report provides significant data and advances to enlight the transcription activation of both maternal and paternal genomes with two waves of activation.

The manuscript is clear and the hypothesis well supported by the data even if few questions remain to be answered by the authors.

- On Figure 1, (Results and Discussion, second paragraph) some of the observed changes of the number of expressed regions are hypothesized to be due to large-scale degradation of maternal transcripts. How the authors do support this hypothesis in the absence of functional tests? What would be the mechanisms and the signal on the transcripts to be recognized for being degraded ? A specific signature?

- Similarly the hypothesis of the stable maternal transcripts (Results and Discussion, seventh paragraph) should be detailed.

- On Figure 2, the two waves of transcriptional activation are detailed and illustrated by the choice of 3 genes for each stage transition. However, the choice of those genes is not clearly justified and why 'only' 3 ones? Some genes associated to defined lineages (epiblast or germ lineage for example) could be also chosen and compared during the establishment of those events.

- As illustrated on Figure 4, some of the signaling pathways – (Results and Discussion, eighth paragraph) were primarily upregulated during the first stages but then down regulated in contrast to the other identified GO terms. Could the authors detail the signaling pathways that emerge in the second wave.

- In the Supplementary file 4, several GO terms are linked to the DNA replication, cell cycle and active proliferation of the cells, with a high p value. Even if it could be expected as the embryo is engaged into an active cleavage and cellularization process, the authors do not mention those facts in the manuscript. Could it be corrected ? Is the embryonic transcription silent for those specific genes?

- Similarly, in Supplementary file 5B, the list of TFs should perhaps be presented completely with the gene names, in particular, the Zn-C2H2 genes as only few of them appear to be expressed (on the 721 identified through HGNC list. More details should be provided to illustrate the specificity – or not – of those first transition.

- As a general question, the repetitive sequences are present in the avian genome as in other genomes and their regulation is also highly dependent on the developmental stages as demonstrated in several species. The authors should at least mention those facts and explain why they excluded them from their analysis.

As a conclusion the manuscript is highly important for the avian field and only few comments have probably to be addressed before being considered for publication.

[Editors’ note: what now follows is the decision letter after the authors submitted for further consideration.]

Thank you for submitting your article "Only the maternal genome is transcribed in chicken zygote upon fertilisation" for consideration by *eLife*. Your article has been reviewed by three peer reviewers, one of whom is a member of our Board of Reviewing Editors, and the evaluation has been overseen by Patricia Wittkopp as the Senior Editor. The reviewers have opted to remain anonymous.

The reviewers have discussed the reviews with one another and the Reviewing Editor has drafted this decision to help you prepare a revised submission.

Summary:

Overall, you will see that two of the reviewers essentially raise the same points as each other. They mainly relate to the clarity of the writing style and how previous work, both from your lab and from others (including other species) is treated. There is some question as to the degree to which the findings described are novel or unusual enough to merit publication in a journal of the calibre of *eLife*. For this reason, we have decided to invite you to submit a final revised version of your paper provided that you can really address these concerns. On balance and in agreement with the reviewers, I feel that the resulting paper could be both more scholarly and shorter, emphasizing the aspects that are truly novel and interesting in a more direct way.

We agree that If this early wave is real, biologically meaningful, RNA polymerase-II driven transcription that only occurs from the maternal genome and not simply noise, this would be a fascinating new discovery. However, with the current presentation of the work, the reviewers were not quite convinced. They felt that the conclusion the authors wish to draw is too important to be presented in this difficult to decipher and experimentally incomplete format.

Essential revisions:

Major re-writing and re-structuring. The literature on the maternal/zygotic transition and differential use of maternal and paternal genomes both in avian and in other species must be incorporated as past work/background and the novel aspects clearly emphasized.

Reviewer #1:

I read the previous version of this paper and I feel that the manuscript has improved. However some problems remain. The main one is that the English can still do with some improvement. Specifically, even the main message of the paper, that the maternal (but not paternal) genome is activated in a first wave of ZGA in chicken, is difficult to extract from the writing style. Here are two examples:

Abstract: "Surprisingly, maternal genome activation was exclusively found in the zygote stage.…". The grammar is very misleading. It should read "Surprisingly, only the maternal genome was found to be activated in the zygote stage.…" (and they could even add, for greater clarity "…, the paternal genome remaining silent until the XXX stage").

Title: "Only the maternal genome is transcribed in chicken zygote upon fertilisation". This is somewhat clearer than the Abstract, but the latter part of the title does not make it clear that this is a transient situation – it sounds as if the paternal genome is never activated. A possible alternative might be: "Zygotic gene activation in chicken occurs in two waves, the first involving only the maternally-derived genome".

I understand that this is a complex situation to describe for non-native English speakers but it is crucial for the message of this paper to come across clearly. More editing is still required throughout the manuscript to improve clarity.

Lastly, I still feel that the novelty of the work does not emerge very clearly from what has already been published by this group and by others. Findings reported in previous work should not be repeated as being new but only referred to with a literature reference. When previously published data were re-analyzed to make new findings, this should be stated explicitly. The methods should not repeat the description of how previously published data were generated but state that they were the same set, reanalyzed and then describe how. Doing these things will probably help to condense the paper substantially but also to focus the main message more clearly.

Overall this will emerge as a short and sharp paper merely describing that during the first wave of ZGA (previously described), only the maternal genome is activated whereas both maternal and paternal genomes are activated during the second wave. This is not a major conceptual advance, but it is an interesting finding whose uniqueness or otherwise may emerge as more studies of ZGA are done in other species in avians and other phyla.

Reviewer #2:

The authors have provided satisfactory replies to the seven points that were mentioned in the initial manuscript mainly by adding new elements such as supplementary data and/or changes in the text.

Reviewer #3:

The major findings are that zygotic transcription in chicks begins in two waves, similar to findings in other organisms, and, more surprisingly, that transcripts detected in the first wave appear to arise exclusively from the maternal genome.

These preliminary findings could form the basis of a more substantial body of work. However, limitations of this study include:

1) This is entirely descriptive work without experiments to address the biological significance of the early wave of transcription, the significance of transcribing from the maternal genome only, or sufficient alternative approaches to validate the proposed early transcription of the maternal genome. Descriptive work is certainly important but, in this case, is perhaps better suited for an archival journal, as the data as presented do not provide a conceptual advance. With respect to the biological relevance of their findings, it should also be noted that Abe et al., 2015, identified promiscuous transcription of many low abundance RNAs in the 1-cell mouse embryo, including transcription of intergenic regions lacking clear promoters (Abe et al., 2015); are the authors here making a similar observation of spurious new transcripts or is there something biologically significant about the early transcripts they report here?

2) Concerns about novelty given prior work on ZGA in avians (chick and quail) and prior work showing an early wave of new transcripts in mice, nematodes, sea urchins, and other organisms. The onset of zygotic transcription in avians has been addressed in at least two other publications that are not mentioned by the authors when they claim that "no detailed investigation of the dynamic transcriptional events occurring at fertilisation in avian species has been reported." This leaves out the work of Nagai et al., who suggested that transcription begins at the 7th to 8th cell division (64-128 cell stage) in chick, based on the levels of actively transcribing RNA polymerase II, and Olszanska et al., who reported that zygotic transcription begins in quail during early cleavage stages. It would help this manuscript to point out these findings explicitly and explain why their findings differ (largely in the identification of the putative early wave of new transcripts).

3) Difficult to read text, written primarily for a highly specialized audience. The Introduction, for example, is a poorly organized, assortment of topics on early gene expression, and also lacks any mention of the major contributions to understanding of zygotic genome activation from work in *Drosophila, Xenopus*, sea urchins, *C. elegans*, and other model organisms, where much of the essential work on ZGA was initially done.

With respect to specific experimental concerns: It would help to provide a more detailed analysis of early gene expression for multiple candidates (especially those expressed in the first wave of transcription), using alternatives to RNA-Seq, as they have done by RT-PCR for a very limited number of genes at a limited number of early stages. In addition, Figure 1C refers to intronic sequences. Are the mature mRNAs corresponding to these intronic sequences expressed? Importantly, what is the level of expression of 1st wave transcripts compared to the 2nd wave of activation? Are they expressed at levels that could have a biological impact? Are these transcripts similar in abundance to the low abundance random expression of unclear significance reported in mice?

[Editors' note: further revisions were requested prior to acceptance, as described below.]

Thank you for resubmitting your work entitled "Zygotic gene activation in the chicken occurs in two waves, the first involving only maternally derived genes" for further consideration at *eLife*. Your revised article has been favorably evaluated by Patricia Wittkopp (Senior Editor) and a Reviewing Editor.

The manuscript has clearly been improved significantly but there are 3 remaining issues that need to be addressed before acceptance, as outlined below:

1) The description of the method (transcriptomics) and how the study in this paper differs from the previous study, which was the subject of criticism by one of the reviewers but clearly also confused the others, is still extremely confusing. It now reads:

"This study was conducted using one type of WGS data and two kinds of WTS data. We declare here to avoid confusion of two different WTS data. Of two types of WTS data, one is bulk embryonic RNA sequencing data that is generated in previous studies (Hwang et al., 2018b, Hwang et al., 2018c) for investigating expression profile of protein coding genes (GSE86592). In this study, we defined this data as "bulked WTS data". Other data is newly generated data for this study and is WTS data for single embryos, which is a descendant of whole genome sequenced samples. Here we defined "single embryonic WTS data" for this dataset. In this study, bulk embryonic RNA sequencing data was reused and analysed, but other analyses were performed."

This is extremely badly written and confusing to the point of being almost incomprehensible. I think the authors are trying to say that while Hwang et al., 2018b, c published RNA-seq data derived from RNA pooled from several embryos, the present study uses new RNA-seq datasets from single embryos. WGS and WTS also needs to be explained more clearly. It also needs to be more explicit in terms of which data are derived from the previously published studies and which are from the new single-embryo RNA-seq. This is very important. A shorter, clearer description will help considerably!

2) As above, the English still needs considerable revision throughout the manuscript.

The first thought that comes to mind is that the company that has been "helping" the authors to polish their English is unable to do so properly either because of their lack of understanding of the science, or because of lack of care. Recurring problems with the use of the definite article, and many other problems with the grammar of the manuscript throughout, suggests that it is the latter and I would strongly advise the authors to find other sources of advice on the language for the future. This is not just a problem that can be solved with an automated spell checker. Whoever advises on the English must take particular care to ensure that the text is absolutely clear. I think this requires the authors to work directly with the advisors. Please have another pass at ensuring clarity and simplicity of the writing.

3) At the same time the length of this article is about 1000 words beyond the limit for a Short Report. Please shorten it to the 2000 word limit. It should not be impossible to do this, especially because at present the English is too convoluted. The paper will greatly benefit from being more punchy, less speculative and more direct.

---

## [Author Response]

[Editors’ note: the author responses to the first round of peer review follow.]

We believe that all raised issues were resolved in the submitted draft. Although we used the RNA-seq data used in previous papers again (Hwang et al., 2018; Hwang et al., 2018,), we performed intron region analysis for the first time to examine pre-mRNA signals that were not previously conducted. A new hypothesis was established through discovery in this initial analysis, and multi-omics data including whole genome sequencing and whole transcriptome sequencing data were newly generated to examine the hypothesis. This newly produced data was used only for this paper except for the data deposit in the repository.

We also revised our manuscript according to the comments and suggestions, and prepared point-by-point responses to the reviewers. Finally, we edited the English in this revised manuscript with two native speakers of an English editing company.

Reviewer #1:[…] I only have one major comment: the paper is very poorly written and the English needs a lot of attention. In places it is almost impossible to understand (even the title makes no sense, as are parts of the Abstract). It is essential that the authors seek help from native English speakers who could help to make this paper readable. I find myself unable to criticise it more deeply because there are several parts which I don't understand.

We are thankful of the comments for improving our manuscript by the reviewer. As reviewer suggested, we checked the revised manuscript with two native English speakers from a professional English editing company.

When re-writing, please be careful to distinguish very clearly between transcription from "maternally-derived genes", and "maternal determinants" (i.e. gene products inherited from the mother, as in anamniotes before ZGA) – because the concept of MBT and late activation of the zygotic genome in many model species is so engrained in the developmental literature, there is some danger than some readers may be confused by these to quite different concepts/findings.

Thanks for this valuable feedback for improving our paper’s readability. In order to distinguish clearly between two concepts, we mentioned about “maternally stored RNAs” in addition to induced transcripts derived by ZGA at the beginning of Introduction(first paragraph).

Reviewer #2:[…] 1) Not enough information is given on the bioinformatics analysis of variants presented for the data in Supplementary file 5E. Allele depth, read depth and SNP quality needs to be presented.

We are thankful to the considerate comments on our manuscript. As the reviewer’s comment, we included the quality control information for detected SNPs in Supplementary file 5B, D (Please find second and fourth taps for WGS and WTS, respectively).

2) How are the 8 gene expression clusters identified in the author's previous paper (Huang, 2018) related to the DEGs identified in this paper?

Single embryonic RNA-seq generated in this study covered oocyte, zygote, and EGK.X stages from each hen. Thus, original clusters in seven patterns can be classified into five clusters (Original cluster 1, 2, 3, 4/5/6, and 7), because of the absence from EGK.I to EGK.VIII. Many genes seem to be included in DEGs, which are in similar pattern to our previous paper (Results and Discussion, fifth paragraph). Also, we provided gene expression information involved in the clusters as Supplementary file 3.

3) My concern is the ability to distinguish between maternal transcripts in the early zygote and de novo RNA transcription. This is premised by the author's statement:'We also investigated primary transcripts throughout early development to obtain definitive evidence of gene activation as the massive alteration of maternally stored RNAs after fertilization may have masked the smaller effects of the 1st activation.'The presence of maternal RNAs is clouding the analysis of de novo transcription. Can the authors identify maternal transcripts through looking at unmapped reads from the 3' end of follicle transcripts and either de novo transcript assembly on these follow by a molecular biology analysis of the transcripts to verify the maternal 3' UTR patterns?

Thanks for your valuable feedback on our study. In this study, we tried to find the definite evidence of 1^st^ wave transcription after fertilisation in chicken zygotes because maternally stored RNAs are enriched in early embryos. We analysed newly expressed intronic region to investigate the definite evidence of de novo transcription after fertilisation, based on whole-transcriptome RNA-seq (revised Figure 1), which is same strategy with previous studies (Lee et al., 2013; Paranjpe et al., 2013, etc.)(Results and Discussion, first paragraph). Furthermore, candidate regions identified in this study was validated using Exon-intron PCR method (Figure 2). As reviewer’s suggestion, the translation of maternally stored mRNA is controlled depending on the difference of 3’ UTR during oocyte maturation (Yang et al., Genes Dev 2017). The diverse 3’UTR pattern between maternally stored and newly expressed transcripts could be found after fertilization. This topic is much interesting, but seems to be little out-side in main aim of this paper.

4) I believe the title is miswritten. 'Only the maternal genome is transcribed in avian zygotes after fertilisation'.

As the reviewer’s suggestion, the title is changed as ‘Only the maternal genome is transcribed in chicken zygote upon fertilisation’.

5) There are several problems with nomenclature. Referring to avian zygotes suggests that several avian species were examined. The authors only examined chicken (Gallus gallus). We do not know if this mechanism is conserved in other bird species.

We greatly agree with the reviewer’s comment. We have used the term chicken zygote to describe our results throughout the revised manuscript.

6) The English needs to be improved. Sections of the paper are difficult to understand.

As reviewer suggested, we checked the revised manuscript with two native English speakers from a professional English editing company.

7) Figure 1A: A time scale is needed. Chicken oviposition requires ~20-24 hours. An approximate stages of EGK should be given.

According to the reviewer’s suggestion, we added the time scale (hours after fertilisation) for each stage in revised Figure 1A.

Reviewer #3:[…] The manuscript is clear and the hypothesis well supported by the data even if few questions remain to be answered by the authors.- On Figure 1, (Results and Discussion, second paragraph) some of the observed changes of the number of expressed regions are hypothesized to be due to large-scale degradation of maternal transcripts. How the authors do support this hypothesis in the absence of functional tests? What would be the mechanisms and the signal on the transcripts to be recognized for being degraded ? A specific signature?

We are thankful of the helpful comments for improving our manuscript by the reviewer. The massive degradation of maternal RNA starts during maturation of oocyte, and continues after fertilisation (Alizadeh et al., Mol Reprod Dev 2005). These degraded genes are appeared to be required for meiosis, but not for early embryonic development. In addition, down-regulated genes after fertilisation were observed in mouse and human early embryos also (Xue et al., 2013). This discussion will be helpful for the potential readers, so we mentioned it in Results and Discussion (second paragraph).

- Similarly the hypothesis of the stable maternal transcripts (Results and Discussion, seventh paragraph) should be detailed.

After 2^nd^ wave of transcriptional activation, the induced genes would be expressed from both alleles in chicken based on our transcriptomic and Sanger sequencing analysis shown in revised Figure 4, and biallelic expression of mammalian orthologues of imprinting genes in chicken embryos validated by a previous study (Frésard et al., 2014). The genes showing only maternal SNP pattern in EGK.X could be considered to be residual maternal transcripts without zygotic expression by 2^nd^ wave. We added a detailed description in Results and Discussion(sixth paragraph).

- On Figure 2, the two waves of transcriptional activation are detailed and illustrated by the choice of 3 genes for each stage transition. However, the choice of those genes is not clearly justified and why 'only' 3 ones? Some genes associated to defined lineages (epiblast or germ lineage for example) could be also chosen and compared during the establishment of those events.

Firstly, we investigated genome-wide intronic expression using whole-transcriptome analysis and observed 1^st^ and 2^nd^ wave of transcriptional activation based on primary transcripts including intronic sequence after fertilisation and EGK.VI (revised Figure 1). Then, the chosen genes are clearly up-regulated intron sequence between oocyte and zygote, and between EGK.III and EGK.VI, shown in Supplementary file 1. Also, these genes are well known to be involved in various biological processing such as transcription factor (*DLX6* [Gitton et al., Development 2011], *ZIC4* [Chervenak et al. Dev Dyn 2013]), lineage segregation and differentiation marker (*GATA2* [Sheng and Stern, Mech Dev 1999], *C8ORF22* [Jiang et al., PLoS Biol 2008]), and signalling-related genes (*WNT11, WNT3A* [van Amerongen and Nusse, Development 2009]). In this regard, we chose these six candidate genes for the investigation of 1^st^ and 2^nd^ wave of transcriptional activation as representatives. We have stated the detailed description in Results and Discussion (third paragraph) of revised manuscript.

- As illustrated on Figure 4, some of the signaling pathways – (Results and Discussion, eighth paragraph) were primarily upregulated during the first stages but then down regulated in contrast to the other identified GO terms. Could the authors detail the signaling pathways that emerge in the second wave.

In this study, we only generated whole-transcriptome sequencing on oocyte, zygote, and EGK.X embryo from each hen. Also, we focused on the maternally expressed functional genes induced by 1^st^ wave, because we covered signalling pathways by 2^nd^ wave in our previous studies (Hwang et al., 2018 and Hwang et al., 2018). According to the previous study, 1^st^ wave-activated Notch, Wnt, and small GTPase signalling are decreased during MZT, while another Wnt ligands such as *WNT8C* and TGF-β signalling are induced by 2^nd^ wave. As the reviewer’s suggestion, we have added related discussion and mentioned our previous studies in Results and Discussion (seventh paragraph).

- In the Supplementary file 4, several GO terms are linked to the DNA replication, cell cycle and active proliferation of the cells, with a high p value. Even if it could be expected as the embryo is engaged into an active cleavage and cellularization process, the authors do not mention those facts in the manuscript. Could it be corrected ? Is the embryonic transcription silent for those specific genes?

In Supplementary file 4A, GO terms involved in active proliferation, such as DNA replication (4.26E-06) and cell cycle (5.62E-04), appeared to be lowest p-value, indicating active cleavage and cellularsation process after fertilisation in chicken.

- Similarly, in Supplementary file 5B, the list of TFs should perhaps be presented completely with the gene names, in particular, the Zn-C2H2 genes as only few of them appear to be expressed (on the 721 identified through HGNC list. More details should be provided to illustrate the specificity – or not – of those first transition.

Our TFs analysis based on gene list in AnimalTFDB including total 817 genes. Among them, ZnF_C2H2 is involved in all three representative patterns (5, 7, and 15 TFs each). Also, the enrichment tests for each pattern showed significantly different *P*-value (3.42E-03, 5.13E-05, and 1.17E-09 each), which means that the TFs containing ZnF_C2H2 domain are involved in early embryogenesis in chicken. As the reviewer’s suggestion, we added whole list of TFs subjected for SMART domain analysis in Supplementary file 5B and the related sentences were mentioned in Results and Discussion (seventh paragraph).

- As a general question, the repetitive sequences are present in the avian genome as in other genomes and their regulation is also highly dependent on the developmental stages as demonstrated in several species. The authors should at least mention those facts and explain why they excluded them from their analysis.

Thank you for your valuable feedback on our study. Our whole-genome sequencing of parents and whole-transcriptome sequencing of their embryos were analysed to identify expressed allele from male or female after fertilisation. However, we believe it is not practically possible to distinguish allelic expression based on repeat sequences in our data, which recently relied on short-read sequencing technology. Also, only long transcripts such as mRNAs and lincRNAs were shown to be induced by 1^st^ wave. We added related sentences in Results and Discussion (sixth paragraph) of revised manuscript.

As a conclusion the manuscript is highly important for the avian field and only few comments have probably to be addressed before being considered for publication.

[Editors' note: the author responses to the re-review follow.]

Reviewer #1:I read the previous version of this paper and I feel that the manuscript has improved. However some problems remain. The main one is that the English can still do with some improvement. Specifically, even the main message of the paper, that the maternal (but not paternal) genome is activated in a first wave of ZGA in chicken, is difficult to extract from the writing style. Here are two examples:Abstract: "Surprisingly, maternal genome activation was exclusively found in the zygote stage.…". The grammar is very misleading. It should read "Surprisingly, only the maternal genome was found to be activated in the zygote stage.…" (and they could even add, for greater clarity "…, the paternal genome remaining silent until the XXX stage").

We are thankful of the comments for improving our manuscript by the reviewer. We modified and added the suggested sentences by the reviewer in Abstract for clarity.

Title: "Only the maternal genome is transcribed in chicken zygote upon fertilisation". This is somewhat clearer than the Abstract, but the latter part of the title does not make it clear that this is a transient situation – it sounds as if the paternal genome is never activated. A possible alternative might be: "Zygotic gene activation in chicken occurs in two waves, the first involving only the maternally-derived genome".

We also agree with reviewer’s comment. According to the reviewer’s suggestion, title was changed to “Zygotic gene activation in the chicken occurs in two waves, the first involving only maternally derived genes”.

I understand that this is a complex situation to describe for non-native English speakers but it is crucial for the message of this paper to come across clearly. More editing is still required throughout the manuscript to improve clarity.

We agree with the reviewer’s opinion. We made extensive modifications in the whole manuscript including Introduction and Results and Discussion. Also, we checked the revised manuscript with two native English speakers from a professional English editing company.

Lastly, I still feel that the novelty of the work does not emerge very clearly from what has already been published by this group and by others. Findings reported in previous work should not be repeated as being new but only referred to with a literature reference. When previously published data were re-analyzed to make new findings, this should be stated explicitly. The methods should not repeat the description of how previously published data were generated but state that they were the same set, reanalyzed and then describe how. Doing these things will probably help to condense the paper substantially but also to focus the main message more clearly.

As the reviewer pointed out, we have resolved an issue that exists throughout the article. In particular, we added the mentions about the limitation on 1^st^ wave ZGA in previous works in avian species in Introduction (first paragraph). In addition, we restructured Materials and methods to describe the analysis used in this study. Also, we added a paragraph about previously generated bulked embryonic whole-transcriptome sequencing (WTS) data and single embryonic WTS data at the beginning of Materials and methods (first paragraph). This paragraph contains a clearer description of the used data.

Reviewer #3:The major findings are that zygotic transcription in chicks begins in two waves, similar to findings in other organisms, and, more surprisingly, that transcripts detected in the first wave appear to arise exclusively from the maternal genome.These preliminary findings could form the basis of a more substantial body of work. However, limitations of this study include:1) This is entirely descriptive work without experiments to address the biological significance of the early wave of transcription, the significance of transcribing from the maternal genome only, or sufficient alternative approaches to validate the proposed early transcription of the maternal genome. Descriptive work is certainly important but, in this case, is perhaps better suited for an archival journal, as the data as presented do not provide a conceptual advance. With respect to the biological relevance of their findings, it should also be noted that Abe et al., 2015, identified promiscuous transcription of many low abundance RNAs in the 1-cell mouse embryo, including transcription of intergenic regions lacking clear promoters (Abe et al., 2015); are the authors here making a similar observation of spurious new transcripts or is there something biologically significant about the early transcripts they report here?

We are thankful of the helpful comment for improving our manuscript by the reviewer. As the reviewer pointed out, the expressed intergenic regions was not changed during pre-ovipositional development in chicken regardless of transcriptional activation (Figure 1—figure supplement 2), unlike the minor ZGA in mammals (Abe et al., 2015). Also, we found that 1^st^ wave-activated maternal genes were functionally enriched in Notch, Wnt, and GTPase signalling (Figure 4—figure supplement 1), which could be involved in early cleavage (Figure 4C), as suggested by previous studies in chicken (Hwang et al., 2018c) and other species (Castanon et al., 2013, Huang et al., 2015, Priess, 2005, Tse et al., 2012, Zhang et al., 2014). We added such sentences in Results and Discussion (subsection “Identification of two waves of ZGA from primary transcript expression measured by intron-spanning mapped reads on bulked WTS data”, last paragraph).

2) Concerns about novelty given prior work on ZGA in avians (chick and quail) and prior work showing an early wave of new transcripts in mice, nematodes, sea urchins, and other organisms. The onset of zygotic transcription in avians has been addressed in at least two other publications that are not mentioned by the authors when they claim that "no detailed investigation of the dynamic transcriptional events occurring at fertilisation in avian species has been reported." This leaves out the work of Nagai et al., who suggested that transcription begins at the 7th to 8th cell division (64-128 cell stage) in chick, based on the levels of actively transcribing RNA polymerase II, and Olszanska et al., who reported that zygotic transcription begins in quail during early cleavage stages. It would help this manuscript to point out these findings explicitly and explain why their findings differ (largely in the identification of the putative early wave of new transcripts).

We also agree with the reviewer’s comment. Previous studies covered ZGA during early cleavage in chicken and quail, but our study is focused on the very first transcription upon fertilization in zygote. We revealed the first wave of ZGA after fertilization prior to early cleavage, based on genome-wide pre-mRNA expression and PCR validation in chicken. For more clarity of our novelty, we added the references you mentioned and related sentences, and modified our description in Introduction (first paragraph).

3) Difficult to read text, written primarily for a highly specialized audience. The Introduction, for example, is a poorly organized, assortment of topics on early gene expression, and also lacks any mention of the major contributions to understanding of zygotic genome activation from work in Drosophila, Xenopus, sea urchins, C. elegans, and other model organisms, where much of the essential work on ZGA was initially done.

We have modified the article to be more straightforward so that a broad range of potential readers can better understand our findings. We also added more examples about ZGA in many species, together with previous studies as references, and made focusing on 1^st^ wave ZGA inIntroduction (first two paragraphs).

With respect to specific experimental concerns: It would help to provide a more detailed analysis of early gene expression for multiple candidates (especially those expressed in the first wave of transcription), using alternatives to RNA-Seq, as they have done by RT-PCR for a very limited number of genes at a limited number of early stages.

As the reviewer’s suggestion, we validated more candidates (revised supplementary file 1) in Figure 2—figure supplement 1using exon-intron RT-PCR and added related sentences in Results and Discussion (subsection “Verification of de novo transcripts using exon-intron reverse transcription-PCR”). In addition, we believe that the two different WTS data we generated are good materials to validate each other. The revised draft explained this fact more clearly. Please see Results subsection “Reaffirmation of the transcriptional activation using WTS data from single embryo”.

In addition, Figure 1C refers to intronic sequences. Are the mature mRNAs corresponding to these intronic sequences expressed? Importantly, what is the level of expression of 1st wave transcripts compared to the 2nd wave of activation? Are they expressed at levels that could have a biological impact? Are these transcripts similar in abundance to the low abundance random expression of unclear significance reported in mice?

We thanks for pointing out an important issues. First, intronic mapped reads represent the expression of pre-mRNA. Many previous studies on ZGA were performed based on this approach to define de novo expression (Abe et al., 2015, Graf et al., 2014, Lee et al., 2013b, Paranjpe et al., 2013).

Second, mean logFC of 1^st^ wave activated genes was 2.187 between oocyte and zygote, and that of 2^nd^ wave activated genes was 2.489 between EGK.III and VI, based on the expression level of pre-mRNAs. This implied that the relative transcriptional activity between two waves were similar and the expression levels could have enough for a biological impact.

Finally, as we mentioned in our first response to reviewer #3, these transcripts seem to be from a genic region which could be involved in cleavage period functionally, not from an intergenic region (Figure 1—figure supplement 2 and Figure 4—figure supplement 1). Although the validation studies in mouse for inefficient splicing and 3’ processing by 1^st^ wave are not able in avian species, these early expression are not shown to be the low abundance in random expression and unclear functions.

* The affiliation #1 and #2 were updated.

* Figure 3—figure supplement 2 was corrected and simplified to present the scheme of single embryo acquisition.

* Key resources table was added at the start of Materials and methods.

* An unnecessary content, Supplementary file 3, was removed in revised draft.

* The custom python script was attached as Source code file 1.

[Editors' note: further revisions were requested prior to acceptance, as described below.]

The manuscript has clearly been improved significantly but there are 3 remaining issues that need to be addressed before acceptance, as outlined below:1) The description of the method (transcriptomics) and how the study in this paper differs from the previous study, which was the subject of criticism by one of the reviewers but clearly also confused the others, is still extremely confusing. It now reads:"This study was conducted using one type of WGS data and two kinds of WTS data. We declare here to avoid confusion of two different WTS data. Of two types of WTS data, one is bulk embryonic RNA sequencing data that is generated in previous studies (Hwang et al., 2018b, Hwang et al., 2018c) for investigating expression profile of protein coding genes (GSE86592). In this study, we defined this data as "bulked WTS data". Other data is newly generated data for this study and is WTS data for single embryos, which is a descendant of whole genome sequenced samples. Here we defined "single embryonic WTS data" for this dataset. In this study, bulk embryonic RNA sequencing data was reused and analysed, but other analyses were performed."This is extremely badly written and confusing to the point of being almost incomprehensible. I think the authors are trying to say that while Hwang et al., 2018b, c published RNA-seq data derived from RNA pooled from several embryos, the present study uses new RNA-seq datasets from single embryos. WGS and WTS also needs to be explained more clearly. It also needs to be more explicit in terms of which data are derived from the previously published studies and which are from the new single-embryo RNA-seq. This is very important. A shorter, clearer description will help considerably!

We are thankful of the helpful comment to improve our manuscript by editors. As editors pointed out, we modified the explanation for datasets, including published WTS (subsection “Identification of differentially expressed regions during early developmental stages of chickens”, last paragraph), WGS (Line 225-238) and newly generated WTS (subsection “Chicken early hybrid embryo preparation, RNA isolation and library preparation for single embryonic WTS data”) in Materials and methods, and subsection “Data availability”, to state used datasets clearly.

2) As above, the English still needs considerable revision throughout the manuscript.The first thought that comes to mind is that the company that has been "helping" the authors to polish their English is unable to do so properly either because of their lack of understanding of the science, or because of lack of care. Recurring problems with the use of the definite article, and many other problems with the grammar of the manuscript throughout, suggests that it is the latter and I would strongly advise the authors to find other sources of advice on the language for the future. This is not just a problem that can be solved with an automated spell checker. Whoever advises on the English must take particular care to ensure that the text is absolutely clear. I think this requires the authors to work directly with the advisors. Please have another pass at ensuring clarity and simplicity of the writing.

As editors’ suggestion, we have checked the revised manuscript from another English editing company.

3) At the same time the length of this article is about 1000 words beyond the limit for a Short Report. Please shorten it to the 2000 word limit. It should not be impossible to do this, especially because at present the English is too convoluted. The paper will greatly benefit from being more punchy, less speculative and more direct.

As editors suggested, we have shortened the main text to 1,866 word count. In this course, we have reduced repetitive contents in Introduction and throughout the manuscript, and have excluded the contents out of the key finding (Figure 4—figure supplement 1B, C and Supplementary file 4B).